# Therapeutic Potential of Bovine Milk-Derived Extracellular Vesicles

**DOI:** 10.3390/ijms25105543

**Published:** 2024-05-19

**Authors:** Madhusha Prasadani, Suranga Kodithuwakku, Georgia Pennarossa, Alireza Fazeli, Tiziana A. L. Brevini

**Affiliations:** 1Institute of Veterinary Medicine and Animal Sciences, Estonian University of Life Sciences, 51006 Tartu, Estonia; madhusha.gamage@emu.ee (M.P.); suranga@emu.ee (S.K.); alireza.fazeli@emu.ee (A.F.); 2Department of Animal Sciences, Faculty of Agriculture, University of Peradeniya, Peradeniya 20400, Sri Lanka; 3Laboratory of Biomedical Embryology and Tissue Engineering, Department of Veterinary Medicine and Animal Sciences, Center for Stem Cell Research, Università degli Studi di Milano, 26900 Lodi, Italy; georgia.pennarossa@unimi.it; 4Department of Pathophysiology, Institute of Biomedicine and Translational Medicine, University of Tartu, 50411 Tartu, Estonia; 5Division of Clinical Medicine, School of Medicine and Population Health, University of Sheffield, Sheffield S10 2SF, UK

**Keywords:** bovine, extracellular vesicles, human health, milk, therapeutics, engineering

## Abstract

Milk is a fundamental component of the human diet, owing to its substantial nutritional content. In addition, milk contains nanoparticles called extracellular vesicles (EVs), which have indicated their potential beneficial roles such as cell-to-cell communication, disease biomarkers, and therapeutics agents. Amidst other types of EVs, milk EVs (MEVs) have their significance due to their high abundance, easy access, and stability in harsh environmental conditions, such as low pH in the gut. There have been plenty of studies conducted to evaluate the therapeutic potential of bovine MEVs over the past few years, and attention has been given to their engineering for drug delivery and targeted therapy. However, there is a gap between the experimental findings available and clinical trials due to the many challenges related to EV isolation, cargo, and the uniformity of the material. This review aims to provide a comprehensive comparison of various techniques for the isolation of MEVs and offers a summary of the therapeutic potential of bovine MEVs described over the last decade, analyzing potential challenges and further applications. Although a number of aspects still need to be further elucidated, the available data point to the role of MEVs as a potential candidate with therapeutics potential, and the supplementation of MEVs would pave the way to understanding their in-depth effects.

## 1. Introduction

Milk is the very first source of food consumed by newborns; thus, it plays a major role in supporting the growth and development of offspring and promoting postnatal health. Milk has been a part of human feed since the pre-historic era due to its highly nutritious cargo and its relatively easy accessibility and low cost. Among the different milk varieties, bovine milk is the most common type consumed by adult humans due to the beneficial effects derived from its various components and a recently growing interest in its bioactive properties, including extracellular vesicles (EVs). Mammalian cells secrete three distinct types of EVs, namely exosomes, micro-vesicles, and apoptotic bodies, which can be distinguished in terms of their content, secretion process, and functions based on traditional classifications. In 2018, for the first time, the International Society of Extracellular Vesicles (ISEV) introduced the Minimal Information for Studies of Extracellular Vesicles guidelines, and a generic term “Extracellular vesicles” was adopted for any type of lipid bilayer particles released by cells and unable to replicate by their own due to the absence of a nucleus [1]. EVs consist of a phospholipid bilayer and an aqueous core. The specific molecules present on the membrane or inside the core serve as biomarkers in their identification. However, the composition may differ according to the original cell type, physiological conditions, and/or environmental conditions. In particular, MEVs are secreted in the mammary glands by different cell types, including mammary gland epithelial cells, adipocytes, fibroblasts, and vascular and immune cells [2]. Bovine milk is the most commonly consumed milk among humans, and it is the cheapest and most accessible source of milk among all milk types. There is evidence that bovine milk was used in traditional medicine to treat various disease conditions [3]. Moreover, MEVs are considered ideal candidates for study due to their therapeutic potential given their tolerance for low-pH, high-temperature conditions, less adverse immune and inflammatory response, and easy absorption.

The present review is based on data presented from peer-reviewed research articles published during the last decade (2014–2024) and is related to the therapeutic potential of bovine MEVs. A comprehensive literature search was performed in PubMed, ScienceDirect, Scopus, and Google Scholar search engines, and 61 peer-reviewed original research articles published in English were considered. In the first part, the different MEV isolation and enrichment strategies in practice are discussed, considering their inherited advantages and disadvantages. The different therapeutic applications of bovine MEVs reported in the literature are then summarized, with a specific focus on the role of MEVs on gut and intestine health, skin and wound healing properties, antioxidant effects, anti-inflammatory and immune-modulatory actions, effects on musculoskeletal health, and antifibrotic and anticancer potentials. At the end, the potential challenges of MEVs’ use as therapeutic agents in practical aspects are discussed to identify the gaps between the experimental and clinical applications.

## 2. Isolation and Enrichment of Bovine MEVs

A universally accepted gold standard method for EV isolation has still not been identified, and different protocols are currently being used. One of the main problems to be solved in MEV extraction is related to milk’s highly heterogeneous composition. Indeed, raw milk contains dead cells, cellular debris, milk fat globules, microbiota, casein, and other proteins, and it is important to remove all those contaminants to enrich the MEV content. The removal of fat and cell debris from unpasteurized raw milk before freezing is advisable in most of the MEV isolation protocols, whereas it can affect the yield and the purity of the resultant EVs [4]. In most studies, the centrifugation of raw milk at speeds generating 2000–3000× *g* has been utilized to separate cream/fat and dead cells from the milk [5]. During this centrifugation, the fat globules form a cream layer on top and can be easily removed. The recent Minimal Information for Studies of Extracellular Vesicles (MISEV) guidelines of 2023 have recommended the storage of milk samples at body temperatures for short-time storage before EV separation and the storage of pre-processed milk (whey) for long-term storage [6].

Around 80% of the total protein in raw milk are caseins that form micelles with spherical shapes and size ranges overlapping with EVs. Physical and chemical pre-processing steps have been identified to remove caseins by centrifugation, acid precipitation, chelation with calcium ions, and aggregation by enzymatic treatments. However, these chemical treatments might have effects on MEV membranes and their downstream applications [5,6,7]. The most commonly used EV isolation techniques are depicted in Figure 1.

The most common method for MEV isolations described in the literature is differential ultracentrifugation, which involves the principle of precipitating and separating components under varying centrifugal forces. A series of centrifugation steps with increasing speeds, starting from 300× *g* to remove apoptotic bodies, cellular debris, and dead cells, up to the highest speed of around 100,000× *g* to precipitate small EVs, are used. The resulting pellet is washed with PBS to remove any contaminants and, subsequently, dissolved in PBS [8]. Even though ultracentrifugation is commonly used in practice, it is a very time- and labor-consuming process, and the high speeds used can cause damage to the MEV membranes. Moreover, ultracentrifugation may not be ideal for EV isolations in complex body fluids such as milk due to the co-precipitation of contaminants [9].

A modified version of ultracentrifugation is density gradient ultracentrifugation, which allows particles of different sizes and densities to settle on different density zones in a single centrifugation step [10]. An inert gradient, such as a sucrose density gradient or an iodixanol density gradient, is commonly used to maintain the integrity of the particles. Time and speed need to be well optimized to achieve a proper yield at the end of the process.

The use of chromatography columns, such as size exclusion chromatography (SEC), generates a comparatively more uniform and pure population of EVs [11]. Based on sample types, different polymers with specific bead sizes can be used. Particles flow under gravity through a porous gel polymer, where the speed of passing depends on their size. More in detail, small particles, such as proteins, enter the gel pores and are retained in the column, while EVs are quickly eluted and passed out first. Although low-density lipoprotein (LDL) and very low-density lipoprotein (VLDL) particles can be co-isolated with MEV fractions, due to their size ranges overlapping with EVs [12], SEC mainly results in a more uniform and pure MEV-enriched population compared to other isolation methods [13,14,15]. Commercially available kits based on the principle of compound polymerization precipitation, such as Exoquick™, are also used to isolate EVs. However, this approach is more suitable for small sample volumes, as the reagents are expensive, and lipoprotein co-precipitation makes it unsuitable for MEV isolation [16]. Additional methods for EV isolation involve the use of immunomagnetic beads, field flow fraction, and microfluid devices, and their pros and cons are listed in Table 1.

It is important to note that each method has its inherent advantages and limitations; therefore, the isolation method needs to be carefully selected depending on the starting material, its availability, and, even more importantly, the downstream applications of the isolated EVs. In addition, the impact on the resulting integrity of EVs is a crucial factor to consider, particularly if the downstream application is focused on studying the functional properties of enriched EVs. A quantitative study of a single EV using a high-resolution atomic force microscope has revealed that the resulting EV morphology and size are different based on the mode of enrichment [32]. Similar to other EV types, MEV integrity and functionality are also affected by the method of isolation [4].

SEC can be considered one of the most preferable modes of EV isolation, since it does not involve complex procedures or instruments. Moreover, damage to the particles is lesser in SEC since they are moving under gravity. However, the major defect of SEC compared to other isolation methods is that it can co-isolate particles in the same size range, since isolation is based on size. Thus, coupling SEC with another chromatography method would be an option to improve the purity and enrich the EV subsets, as a recent report highlights that the double-SEC method even improves the purity [33]. Density gradient ultracentrifugation is more reliable than conventional ultracentrifugation, since particles are separated based on the solution density and have a lesser effect on the EV integrity. Apart from the described conventional methods of EV isolation, new techniques involving microfluid devices and flow filtration are currently emerging as promising strategies, although facilities are required.

However, it is wiser to use a combination of two or more of the above-described methods to eliminate the limitations of a single method resulting in a more uniform population of MEVs [23]. The recent MISEV 2023 guidelines recommend positioning new EV isolation methods on a recovery/yield versus specificity grid to facilitate the comparison with existing methods and the usage of combined methods [6].

## 3. Different Therapeutic Aspects of MEVs

There are numerous reports on the beneficial roles of MEVs in in vitro and in vivo studies in the literature. Figure 2 depicts a summary of the possible beneficial roles of MEVs on the hypothesized organs.

### 3.1. The Role of MEVs in Gut and Intestine Health

Since milk is a crucial part of the diet, research on the MEV effects on intestine and gut health has become an emerging field during the last decade. Previous studies have demonstrated that EVs and their labile cargo are not degraded in the harsh digestive environment, but, conversely, they are selectively uptaken by intestinal cells via endocytosis, making MEVs an ideal candidate for therapeutic applications [34,35]. Recent studies have shown that although digestive enzymes and bile can affect the quantity of MEVs at the end of the digestion phase, certain MEV types remain intact with the cargo [36]. Moreover, dietary MEVs can cross the placenta and increase embryo survival rates in mice models [37]. MEVs contain bioactive miRNAs that are present in sufficient amounts to induce changes in gene expression levels within the target organs [38].

In general, EVs play a major pathophysiological role in the modulation and proper functioning of the intestine by preventing inflammatory processes, altering the gut microbiota, and limiting and repairing intestine epithelial damage. In bovines, they can improve the intestine epithelial cell proliferation and restore villus architecture and the intestinal barrier, thus facilitating a higher nutrient absorption during malnutrition [39] and/or leaky gut conditions [40]. Both cow and yak MEVs have the potential to reduce lipopolysaccharide-induced intestine inflammation in IEC-6 cells via the activation of the PI3K/AKT/C3 pathway [41]. In addition, yak milk EVs were found to promote intestinal epithelial cell proliferation in hypoxic conditions through their mRNA cargo [42,43]. However, it must be noted that these effects may differ depending on the specific genetic and immune responses distinctive to the animals from which the EVs were derived [44]. Elucidating the variability is complicated due to various unknown factors and insufficient knowledge about the molecules and pathways involved in the protective effects.

Bovine MEVs can reduce ulcerative colitis induced by dextran sodium sulfate (DSS) by removing the reactive oxygen species and immune cytokine regulations [45]. Interestingly, detailed studies have revealed that MEV mRNA cargo upregulates TGFβ1 protein levels in colon tissues [46] and inhibits the pro-inflammatory cytokines and chemokine expressions in DSS and tamoxifen-induced colitis mice models [47,48]. Furthermore, MEVs can correct bacterial dysbiosis in the colon [49,50], improving intestinal epithelial cell proliferation and extracellular matrix formation [51]. Since MEVs can alter the composition of the gut microbiota and their metabolite production, in turn, gut microbes can activate MEV signaling, suggesting a potential prebiotic role for MEVs [52,53]. Recent studies have indicated that bovine colostrum-derived EVs can regulate the host–pathogen interaction [54] and exert a specific antimicrobial effect against pathogens, such as E. Coli, by changing the cGAS/STING pathway, which is directly involved in the development of inflammation [55]. Thus, colostrum EVs can be used as an alternative antibiotic source for the treatment of calf neonatal diarrhea conditions.

MEVs have been proven to improve intestinal health, specifically during necrotizing enterocolitis (NEC), a fatal disease for premature infants. Recently, it has been demonstrated that MEVs exert a prophylactic effect on the NEC model in vitro via the induction of goblet cell functions, which are responsible for the mucous production that prevents intestinal injuries [56]. In addition, parallel studies have revealed that, when orally administered, bovine MEVs can reduce NEC-related lung injuries via the regulation of the NF-κB signaling pathway, eventually reducing inflammatory factors [57].

The most convenient method of using MEVs as a therapeutic agent is via oral administration. After passing the gastrointestinal barrier, the MEVs are absorbed by the intestine cells and transported to the liver and subsequently to the other organs. In-depth studies on this transportation to targeted organs will shed more light on understanding the mechanisms and mode of action of MEVs. A summary of the therapeutic effects of MEVs on gut and intestine health is given in Table 2.

### 3.2. Effects on Skin and Wound Healing

In recent years, cow MEVs have emerged as a promising avenue for enhancing the efficacy of skin-whitening agents. This is primarily attributed to their ability to downregulate the expression of melanogenesis-associated genes and tyrosinase activity in melanocytes and melanoma cells. The resulting reduction in melanin content offers a safe and low-toxicity approach to skin whitening that is highly desirable for use in the cosmeceutical industry. This effect appears to be related to the relatively high content of miR-2478 in MEV cargo [58]. Another proven beneficial effect of MEVs is their wound healing potential and scar-free treatment effects. They are indeed able to activate different mechanisms in human stem cells, such as wound healing and pro-angiogenic pathways, thus stimulating collagen synthesis and inducing the secretion of various growth factors.

MEVs also control the inflammatory response by regulating TGFβ isoforms, promote both skin cell proliferation and migration, and display the ability to reduce the expression levels of major inflammatory mediator mRNAs in a dose-dependent manner [59]. These properties have been demonstrated when MEVs were introduced as antiaging ingredients in skincare formulations and were able to induce an increase in moisture retention and anti-wrinkling effects in both in vitro as well as in vivo trials, carried out by Lu et al., in human skin [60]. Notably, the lack of sensitization reactions and irritational responses among the 31 female individuals participating in the study was reported. On the other hand, the precise mechanism used by MEVs to penetrate the epidermis barrier is still to be elucidated. In addition to those direct effects, the engineering of MEVs with polydopamine nanoparticles or hydrogels has been shown to increase the wound healing process by facilitating both their localized retention and slow release, boosting PI3-AKT-mediated angiogenic effects similar to what was previously reported for mesenchymal stem cell (MSC)-derived EVs [61]. Interestingly, bovine MEVs engineered with miR-31-5p and siRNA-KEAP1 accelerated healing effects in diabetes-dependent wounds and ulcers [62,63].

In general, the wound healing process involves four overlapping and inter-related stages that comprise hemostasis, inflammation, proliferation, and remodeling [64]. Based on the available data, MEVs can be considered as an ideal therapeutic agent in the wound healing process, given their ability to reduce inflammation and increase cell proliferation. A summary of the therapeutic potential of MEVs related to their effects on skin and wound healing properties is shown in Table 3.

### 3.3. Antioxidant Properties

The imbalance between the production and accumulation of oxygen reactive species (ROS) and the body’s ability to counteract antioxidants’ destructive impacts leads to a condition called oxidative stress [65]. The higher levels of ROS may cause harmful effects on the main vital cellular structures, such as proteins, lipids, and nucleic acids. Oxidative stress, therefore, is the progress of many disease conditions, such as cancer, metabolic disorders, cardiovascular pathologies, gastroduodenal ulcers, inflammatory bowel disease, and gastrointestinal malignancies [66,67]. In this perspective, it is interesting to note that bovine MEVs have been previously demonstrated to exert protective effects against H2O2-induced oxidative stress in IEC-6 cells [68]. Pre-treatment with MEVs increases cell viability and antioxidant enzyme levels, such as superoxide dismutase (SOD) and glutathione peroxidase (GPX), while reducing the ROS, lactate dehydrogenase (LDH), and malondialdehyde (MDA) levels, suggesting MEVs have protective effects on cell membranes exposed to oxidative stress. The downregulation of antioxidant protein-coding genes, Nrf2 and Ho1 expression levels, coupled with increased miR-146a and miR-155 transcription, is considered one of the possible mechanisms.

Furthermore, MEV pre-treatment also affected purine and energy metabolism and influenced the activity of key enzymes involved in nucleotide catabolism and ROS production, such as adenosine deaminase (ADA) and xanthine oxidase (XOD) [69]. Altogether, these data demonstrate the key role of MEVs in protecting the intestinal barrier from ROS’s detrimental effects, thus preventing disease conditions such as ulcerative colitis, colon cancer, and ulcers [70]. A summary of the therapeutic effects of MEVs antioxidant properties is given in Table 4.

### 3.4. Anti-Inflammatory and Immunomodulatory Effects

The immune system is a vital organization inside the body, preventing diseases and supporting growth and development. Human MEVs can transfer their genetic information from the mother to the offspring, improving the immunity of the newborn [71,72]. In agreement with this, MEV metabolite and pathway analysis revealed the presence of key molecules and pathways directly involved in inflammatory and immune modulation processes [73]. Similarly, transcriptomic analysis of MEVs demonstrated the presence of RNA cargo related to immunomodulation, such as miR-151, miR-148, and miR-21 [74]. Also, in bovine species, the colostrum contains significantly higher amounts of proteins that are directly related to the innate immunity of the calves [75]. This feature is maintained in the milk, where MEVs increase the viability and proliferation of macrophages (*RAW 264.7* cells) against cisplatin drug-induced cytotoxicity [76] and lowered the lipopolysaccharide (LPS)-induced NF-κB pathway both under normal [77] and hypoxic conditions [78].

Immune-related miRNAs contained in MEVs are very stable, since they are detected in commercially available milk, which is subjected to pasteurization. Indeed, in vitro studies demonstrated that EVs isolated from pasteurized milk interact with immune cells and regulate T-cell differentiation through the TGF-β pathway [79]. This ability is comparatively low in UHT and freeze-dried-milk powder compared to pasteurized milk, since MEV-fed mice exposed to agriculture dust developed an immune response towards lung inflammation that varied depending on the source of the EVs [80,81,82]. In-depth in vivo studies need to be carried out to find the exact mechanisms involved. A summary of the results of the anti-inflammatory and immunomodulation effects of MEVs is given in Table 5.

### 3.5. Effects on Musculoskeletal Health

Milk consumption has beneficial effects on bone and muscular health, which is a well-known fact and is due to its highly nutritious components, such as calcium, phosphorus, Vitamin D, Vitamin K, proteins, and minerals, such as Mg. On the other hand, the effects of MEVs on muscular health remain unclear, and there exist different perspectives and findings in the literature. Hydrolyzed whey protein-derived EVs and MEVs both cause an increase in skeletal muscle protein synthesis and diameter in C2C12 myotubes, suggesting a possible anabolic effect [83,84]. In contrast to these findings, an in vivo study carried out in a rodent model revealed no significant effects on forearm grip strength or amino acid levels when MEVs were used as a dietary supplement [85]. These controversial results indicate that further studies are mandatory to better elucidate MEV’s role in muscular health.

Clear evidence, in contrast, supports the prophylactic and therapeutic effects of the bovine MEVs in bone and cartilage health. The oral supplementation of MEVs reduced the development of autoimmune arthritis and decreased bone marrow inflammation in a mouse model [86]. Further studies supported the role of bovine colostrum-derived EVs as a prophylactic agent to prevent osteoporosis, promote osteogenesis [87,88], and modify gut microbiota [89]. The key players in bone and cartilage health are mRNAs, such as miR-30a, miR-92a, and miR-21 [90], which support MEVs’ protective activity against bone loss both in arthritis and obesity-induced mouse models via the RANKL/OPG system, which is a balance between bone formation and resorption [91], as well as alveolar bone loss [92]. In addition, recent studies revealed that MEVs induce bone formation, upregulating the transcription of the osteogenic genes GJA1 and AP3B1 [93]. Besides these beneficial effects, however, Oliveira et al. reported that, while MEVs induced a fast osteoblast differentiation, leading to rapid bone formation, this resulted in an impaired bone matrix deposition which reduced the bone quality [94]. A summary of the effects of MEVs on musculoskeletal health is given in Table 6.

### 3.6. Antifibrotic Activity

Fibrosis is the excessive accumulation of the extracellular matrix component proteins, such as collagens, fibronectins, and smooth muscle actins, ultimately leading to organ dysfunctions [95]. Bovine MEVs were uptaken by hepatic stellate cells (HSC), which are the cells that secrete pro-fibrotic bioactive molecules in liver injury and inhibit their proliferation via the overexpression of miR-148, in a mice model [96]. Since HSCs are responsible for the production of extracellular matrix proteins, the MEV-dependent inhibitory effect on their proliferation ultimately results in a decreased deposition of extracellular matrix proteins with a general antifibrotic effect of MEVs. A similar effect was also observed in fibrosis induced by isoproterenol (ISO) in rat models, as well as in HUVEC cells, indicating MEVs’ ability to counteract cardiac fibrosis [97]. Altogether, these preliminary observations point to potential antifibrotic properties exerted by MEVs, although the exact molecular mechanism/s behind this effect is still to be elucidated. A summary of the antifibrotic effects of MEVs is given in Table 7.

### 3.7. Anticancer Properties

Bovine MEVs have had their anticancer potential proven, as have the beneficial effects of cross-species EVs in cancer therapy. Many studies have been conducted on the efficacy of MEVs as anticancer drug-delivery cargo, due to their biological and physical stability in harsh environmental conditions, scalability in bulk productions, versatility, and low immunological response from the body [98,99]. MEVs can decrease the proliferation of neuroblastoma cells, which are a type of pediatric cancer, thus indicating their anticancer potential [100]. Moreover, the combined therapy of MEVs with doxorubicin has induced apoptosis in cancer cells, while increasing their chemosensitivity [101]. Interestingly enough, MEV membranes need to be intact to ensure anticancer action. Indeed, it has been demonstrated that cancer cell proliferation is not reduced when MEV membranes are sonicated [101].

A further intriguing observation is related to MEVs’ ability to induce senescence in primary intestine tumors, while, at the same time, augmenting metastasis and inducing cancer progression in other organs [102]. More in detail, it was demonstrated that EVs secreted by cells that were senescent after MEV exposure were shown to induce the epithelial–mesenchymal transition (EMT) and favor pro-metastatic processes. On the other hand, when MEVs were orally administrated after the surgical removal of the primary tumor, metastasis was attenuated. Therefore, the dosage, timing of MEV administration, and identification of the specific MEV molecular cargo responsible for both anticancer properties and metastasis acceleration are all crucial factors in better understanding MEV’s double-edged-sword role in cancer treatments, since the time of administration determines whether the role is beneficial or adverse. A summary of the findings related to the anticancer effects of MEVs is given in Table 8.

## 4. Challenges of Using MEVs as Therapeutic Agents

Considering the above findings from the last decade, MEVs appear to be very promising therapeutic agents. However, gaps exist between the experimental findings related to their therapeutic applications and clinical trials. Hence, we wish to discuss some of the key challenges and possible ways to overcome them.

### 4.1. Lack of a Standardized Isolation Method

One of the main challenges of using MEVs as therapeutics is the lack of a proper standardized isolation method, since this is a critical step that can damage or alter EV integrity and batch-to-batch variations, thus affecting downstream applications. ISEV has recently formulated a “Milk task force” to collect and share MEV-related findings and to provide sets of guidelines for planning, conducting, and reporting on future MEV-related experiments (https://www.isev.org/milk-task-force (accessed on 29 January 2024). MEVs have been successfully isolated from different milk sources such as raw milk, pasteurized milk, and lyophilized powders; however, the quality and quantity of the extracted MEVs are not comparable. In addition, raw MEV composition changes depending on different genetic, health, and environmental conditions, management practices, and lactation-curve stages [103,104]. The presence of contaminants, including casein micelles, lipoproteins, and cellular debris, can significantly impact downstream applications by introducing unwanted variability and potentially distorting results. All these factors must be taken into consideration to define specific standardized isolation protocols.

### 4.2. Scalability and Cost-Effectiveness

More practical and reliable standard isolation methods are required for scaling up the production of MEVs. As we previously discussed, SEC can be identified as a more promising isolation method with a lesser effect on MEV integrity and function; however, it is not a very efficient method in terms of MEV mass production for therapeutic applications. Combining flow fractionation techniques and flow filtration methods with SEC would be more a promising method in the bulk production of MEVs with higher purity. However, maintaining the sterility and the frequent replacement of filters are essential, since the latter filters can get easily clogged, disturbing the filtration process and promoting bacterial growth. The benefits, including the cost-effectiveness of MEV-derived therapeutics, must exceed the benefits of existing synthetic drugs to have more acceptance. Collaborative aspects can be taken into consideration when finding the starting materials for MEV production to maximize the available sources and avoid wastage. Whey is a byproduct in the cheese industry, and hence it can be used to isolate MEVs. Moreover, improving streamlined workflow with automation and semi-automation steps, investing in further research and development, collaborations between the industry and academic partners, and focusing on the stability of MEVs during storage and transportation would benefit the scaling up of production, with the high purity that is needed in the GMP processes of biological manufacturing.

### 4.3. Storage and the Stability of MEVs in Different Storage Conditions

Maintaining the structural integrity and inherent characteristics of MEVs during extended storage periods remains a significant challenge in the field. The key factors that can affect the physiological and biological characteristics of EVs are temperature, pH, and preservation techniques. There is little evidence that cow and human milk can be stored at −80 °C for 28 days with a minimal loss in physiological characteristics, and in short-term storage at 4 °C for when immediate centrifuge is not possible [105]. However, the long-term storage of unprocessed milk at −80 °C and the effects of repeat freeze–thaw cycles are still controversial. In addition, experiments have been conducted to improve the storage duration with preservation techniques such as cryopreservation, freeze drying, and spray drying. Commonly used cryoprotectants that can penetrate include DMSO and glycerol, while inert sugars such as trehalose and sucrose can be used as non-penetrating cryopreservative agents. Alternatively, lyophilization and spray drying can be improved as more promising preservative techniques, since the resultant EVs can be stored and transported at room temperature.

### 4.4. Potential Risks and Safety Considerations When Using MEVs as Therapeutic Agents

Some of the proteins present in raw milk might be potential allergens to consumers. It is crucial to consider this factor when selecting an isolation method for developing MEVs as therapeutic agents. Moreover, MEVs may trigger an immune response in humans, leading to adverse reactions or immune-related complications due to their recognition as foreign entities by the immune system. There is a potential risk that MEVs may interact with various cellular and tissue targets in an organism, leading to unintended physiological disruptions or negative effects on the recipients. An in vivo study on the anticancer properties of MEVs reported that MEVs have the risk of increasing metastasis, depending on the time of treatment [102]. Therefore, a careful evaluation and understanding of the individual components contained in MEVs, specific biological roles in disease conditions, and specific mechanisms involved must be considered to elucidate and rule out any possible undesirable effects. Another key factor that needs to be considered seriously is the health conditions of the cows, where certain infections (viral and bacterial), and drug residues, specifically antimicrobials, can be enriched with EVs. A promising option to consider would be the in vitro production of synthetic milk [106], where greater uniformity can be achieved and the concerns about potential allergens would be diminished. However, it is necessary to study whether the effect of the MEVs obtained from raw milk and synthetic milk would be the same.

### 4.5. Ethical Considerations of Using MEVs as Therapeutic Agents

Since MEVs are derived from animals, there might be ethical dilemmas and concerns from the animal welfare perspective. Ensuring the well-being of cows, implementing sound management practices for MEV production, and equitably distributing the benefits of therapeutic drugs derived from MEVs will increase public confidence in MEVs in therapeutics. Moreover, it is crucial to mitigate risks and ensure safety by conducting thorough preclinical assessments, implementing robust quality control measures, and following regulatory guidelines when advancing MEV-based therapeutics to clinical translation.

## 5. Future Directions of the MEV Research Field

In-depth studies of the molecular mechanisms of biogenesis and the secretion of MEVs, and crucial study on the cargo of MEVs, will facilitate the regulation of their production and release, as well as potential targets for therapeutic intervention. Advanced techniques such as single-EV analysis and imaging can help us to better understand MEVs’ characteristics, facilitating advanced aspects such as MEV engineering, since they possess great potential as natural drug carriers, owing to their biocompatibility and immunogenicity. In addition, recent studies have shown that MEVs have the potential to replace antibiotics, as they can effectively combat bacterial infections [107].The future of the MEV research field will rely on maintaining standard isolation methods, implementing rules and regulations for clinical trials involving MEVs as therapeutics, encouraging interdisciplinary collaborations, and comprehending the role of MEVs in disease diagnostics.

## 6. Conclusions

When considering all the evidence-based outcomes of research carried out during the last decade, it becomes clear that MEVs have the potential to be therapeutic agents for a vast variety of disease conditions. However, considering the major challenges and practical usage, it is wiser to introduce MEVs as a supplement with added bioactive properties, rather than as a direct therapeutic drug, until more clinical studies are conducted. The introduction of regulations, standards, and legislation for the production, experimentation, and usage of MEVs would help to reduce the gap between the research outcomes and clinical trials.

## Figures and Tables

**Figure 1 ijms-25-05543-f001:**
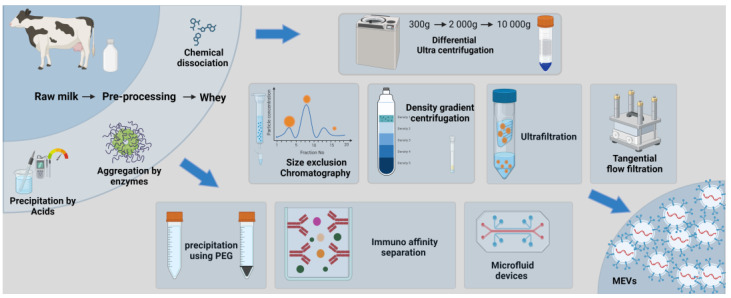
Different EV enrichment methods.

**Figure 2 ijms-25-05543-f002:**
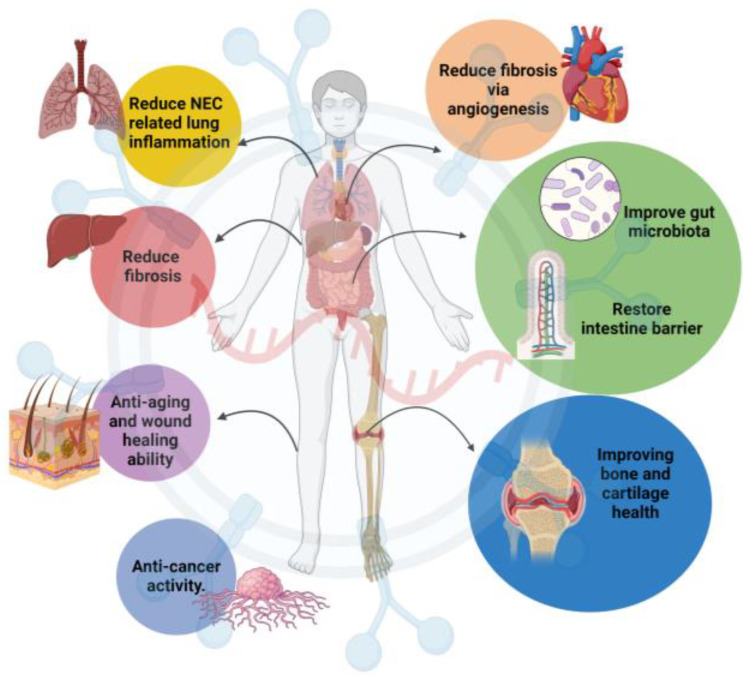
Potential beneficial roles of MEVs. Functions of MEVs in gut- and intestine-related disorders, reduction in fibrosis in heart and lungs, reduction in inflammation, wound healing properties, and improvement in bone and cartilage health.

**Table 1 ijms-25-05543-t001:** Pros and cons of different EV isolation methods.

Method	Advantages	Disadvantages	References
Differential centrifugation/ ultracentrifugation	Low cost Results in a higher number of particles	Damage to the EVs Co precipitation of contaminants Requires a larger starting volume	[17,18,19]
Density gradient centrifugation	Results a comparatively pure population than the ultracentrifugation High separation efficiency Does not affect the integrity of the particles	Low productivity Time consuming Preparation of solutions needs time and a complex procedure Involves expensive instruments	[20,21]
Immunomagnetic beads	Specific subpopulations can be separated based on the expression of specific EV markers regardless of the size	Separating the EVs from the magnetic beads is difficult Expensive method Not suitable for large sample volumes	[22,23]
Commercially available polymer-based precipitation kits	Not labor-intensive Simple and faster method Can be used for small sample volumes High recovery rate More suitable for isolating sEVs	Formation of aggregates Co precipitation of lipoproteins Expensive	[16,24]
Size exclusion chromatography	EVs retain their functionality and integrity High purity Uniform in size Easy to setup Can be used for low volumes	Low yield Dilution of the EV population Not suitable for higher sample volumes	[11,15]
Ultrafiltration	Simplicity of the procedure Can be used for low sample volumes	Deformation of the particles Loss of particles due to the absorption of the membranes Varying efficiency of filtering due to the clogging of the pores in the membranes	[25,26]
Field-flow fractionation	Suitable for scalable production Retaining of particle integrity	Complex instrumentation setup Loss of samples	[27,28,29]
Tangential flow filtration	Suitable for scalable production Reduced sample contamination Recovery efficiency is high	Complex instrumentation setup Frequent changes to the membranes are required	[9,25]
Microfluidic devices	Suitable for small sample volumes Targeted isolation of EVs resulting in a homogenous population	Low recovery rate Needs to be optimized and expensive equipment is involved	[30,31]

**Table 2 ijms-25-05543-t002:** Summary of the therapeutics potential of MEVs related to gut and intestine health.

	Model	Source of MEV	Dosage	References
In vitro studies	IEC-6 cells	Yak and cow milk	50–200 ng/μL of exosome protein	[40]
	IEC-6 cells	Yak milk	50 nM of bta-miR-34a miRNA in exosomes	[41]
	IEC-6 cells	Yak and cow milk	120–240 ng/μL of exosome protein	[42]
	Caco-2 cells	Cow milk and colostrum	0.001–0.625 μg/μL of exosome protein	[43]
	NCM460 and RAW264.7 cells	Cow milk colostrum	0.1–1 mg/mL of exosome protein	[44]
	IPEC-J2 cells	Cow milk colostrum	0.015–150 μg of exosome protein	[55]
	LS174T cells	Cow milk	0.1 μg/μL of exosome proteins	[56]
In vivo studies	C57BL/6J mice	Cow milk	4.83 × 10^6^ milk EVs/g of BW	[38]
	C57BL/6J mice	Cow milk	0.6–3.0 mg/kg BW	[39]
	Balb/c mice	Cow milk	50 mg/kg BW	[45]
	C57BL/6 mice	Cow milk	3.0 × 10^9^ particles/g/BW	[46]
	C57BL/6 mice	Cow milk	1 mg protein/mL BW	[47]
	C57BL/6 mice	Cow milk	0.6–3.0 mg per BW per day	[48]
	C57BL/6 mice	Cow milk	1.5 × 10^8^–1.5 × 10^9^ particles/g BW	[49]
	C57BL/6J mice	Skimmed cow milk	EVs from 10 mL of milk	[51]
	C57BL/6 mice	Cow milk	2 × 10^12^ exosomes/mL	[52]
	BALB/c mice	Cow milk	1 mg/mL/per day	[50]
	C57BL/6 mice	Cow milk	0.3–1.2 mg/kg BW	[53]
	C57BL/6 mice	Cow milk	1 μg/μL/gavage feed	[56]
	C57BL/6 mice	Cow milk	1 μg/μL/feed	[57]

**Table 3 ijms-25-05543-t003:** Summary of the therapeutic potential of MEVs related to effects on skin and wound healing properties.

	Model	Source of MEV	Dosage	Reference
In vitro studies	Mouse melanoma B16F10 cells and human melanoma MNT-1 cells	Cow milk	20 and 50 µg/mL	[58]
	RAW264.7 cells IEC-18 cells	Cow milk	1 × 10^8^–1 × 10^10^ particles/well	[59]
	CCC-ESF-1 and HaCaT cells	Cow milk	6.25–50 µg/mL	[60]
	3T3 cells (mouse dermal fibroblasts) and HUVEC cells	Cow milk	1.0 μg/μL per well	[61]
	HUVEC cells	Cow milk	4 μg per well	[63]
In vivo studies	C57 BL/6J mice	Cow milk	1.0 μg/μL per dose	[61]
	BALB/c mice	Cow milk	1.0 μg/μL of miR-31-5p-loaded MEVs	[62]
	C57BL/6 mice	Cow milk	2 μg/wound	[63]
Clinical study	Human trial	Cow milk	60 μg/mL per twice a day	[60]

**Table 4 ijms-25-05543-t004:** Summary of therapeutic potential of MEVs related to antioxidant properties.

	Model	Source of MEV	Dosage	Reference
In vitro studies	IEC-6 cells	Cow milk (skimmed)	50–800 µg/mL	[68]
	IEC-6 cells	Cow milk (skimmed)	50–800 µg/mL	[69]

**Table 5 ijms-25-05543-t005:** Summary of anti-inflammatory and immunomodulation effects of MEVs.

	Model	Source of MEV	Dosage	Reference
In vitro studies	RAW 264.7 cell	Cow milk (skimmed)	15–60 µg/mL	[76]
	RAW264.7 cells	Cow milk	10–100 µg/mL	[77]
	RAW 264.7 cells	Cow milk (skimmed)	100 and 200 μg/mL	[78]
	Native splenic T cells	Cow milk (semi-skimmed)	400 μg/mL	[79]
	Human PBMCs	Cow milk	10–100 µg/mL	[80]
	MH-S murine alveolar macrophage cell line	Cow milk (semi-skimmed)	24.8 × 10^6^ exosomes per well	[81]
	RAW264.7 cells	Fresh milk, pasteurized milk, UHT milk, freeze-dried powder, and organic milk powder).	1 × 10^6^ EVs/100 µL of media	[82]
In vivo studies	C57BL/6 mice	Cow milk (semi-skimmed)	Feeds were formulated to achieve 10% of total calories from milk	[81]

**Table 6 ijms-25-05543-t006:** Summary of musculoskeletal health effects of MEVs.

	Model	Source of MEV	Dosage	Reference
In vitro studies	C_2_C_12_ myoblast cells	Hydrolyzed whey powder	1–10 mg/mL	[83]
	Saos cells	Cow milk	100–1000 µg/mL	[87]
	MC3T3-E1 and RAW 264.7 cells	Cow milk	20 µg/mL	[88]
	MC3T3-E1 preosteoclast cells	Cow milk colostrum	20–500 ng/mL	[89]
	MLO-Y4 osteocytes	Cow milk (semi-skimmed)	10–100 µg/mL	[91]
In vivo studies	Fisher 344 rats	Cow milk	5–15 µg/µL	[84]
	C57BL/6 mice	Cow milk	Diet included 0.5 L of milk	[85]
	IL-1Ra−/−mice and DBA/1J mice	Cow milk (semi-skimmed)	4 × 10^6^–28 × 10^6^ particles	[86]
	Sprague Dawley rats	Cow milk	0.5–50 mg/BW/dose	[87]
	C57BL/6 mice	Cow milk (semi-skimmed)	14.3 × 10^6^ particles/mL	[92]
	C57BL/6 mice	Cow milk	1.2 μg EVs	[93]
	DBA/1J mice	Cow milk (semi-skimmed)	4.7 × 10^6^/mL or 14.3 × 10^6^/mL	[94]

**Table 7 ijms-25-05543-t007:** Summary of antifibrotic effects of MEVs.

	Model	Source of MEV	Dosage	Reference
In vitro studies	HUVEC cells	Cow milk	100 µg/mL	[97]
In vivo studies	BALB/c mice	Cow milk	13 mg/kg BW	[96]
	Sprague Dawley rats	Cow milk	600 μg of MEV	[97]

**Table 8 ijms-25-05543-t008:** Summary of anticancer effects of MEVs.

	Model	Source of MEV	Dosage	Reference
In vitro studies	Human lung cancer (A549 and H1299), breast cancer (MDA-MB-231, MCF7), pancreatic (PANC1, Mia PaCa2), Prostate (PC3, DU145), colon (HCT116), and ovarian (OVCA432) cells	Cow milk	50 µg/mL of EV protein	[100]
	SK-N-BE2 NBL and C26 colon cancer cells	Cow milk	100 µg/mL	[101]
	SW620 colorectal cancer cells	Cow milk	20 µg/mL	[102]
In vivo studies	Athymic nude mice	Cow milk and colostrum	25 mg protein/kg BW	[98]
	Athymic nude mice	Cow milk	50 mg protein/kg BW	[99]
	Balb/c mice	Cow milk	25 mg/kg	[102]

## Data Availability

No new data were generated for this article.

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
