# Peer review of "Therapeutic Potential of Bovine Milk-Derived Extracellular Vesicles"

_ijms, 2024, doi:10.3390/ijms25105543_

Round 1

Reviewer 1 Report

Comments and Suggestions for Authors

A manuscript submitted for review on the topic: "Therapeutic potential of bovine milk-derived extracellular vesicles" presents the isolation possibilities and the various beneficial properties of extracellular vesicles from cow's milk. In this aspect, the overview is interesting because there are not many similar articles in the literature that summarize the properties of cow's milk in this way.

I have the following notes for the authors:

At the end of each paragraph, it is good to present in the form of a table the amount of milk as a daily dose or presented in another way (week, month) and which breed of cattle, as well as the author who describes it. In this way, the reader will be guided as to what the necessary amounts of intake would be and from which breed.

The notes made are for recommendation only and do not detract from the value of the submitted manuscript, which in my opinion can be accepted for publication.

Author Response

A manuscript submitted for review on the topic: "Therapeutic potential of bovine milk-derived extracellular vesicles" presents the isolation possibilities and the various beneficial properties of extracellular vesicles from cow's milk. In this aspect, the overview is interesting because there are not many similar articles in the literature that summarize the properties of cow's milk in this way.

I have the following notes for the authors:

At the end of each paragraph, it is good to present in the form of a table the amount of milk as a daily dose or presented in another way (week, month) and which breed of cattle, as well as the author who describes it. In this way, the reader will be guided as to what the necessary amounts of intake would be and from which breed.

The notes made are for recommendation only and do not detract from the value of the submitted manuscript, which in my opinion can be accepted for publication.

As suggested by the reviewer, a table including the study model, the source of MEVs, MEV dosages and references were included at the end of each paragraph. A specific cattle breed and amount of milk cannot be included since most of the MEVs were isolated from commercially available milk in the market and the exact starting volume used to isolate MEVs was not included in every reference article. Only the available data were summarized. Please see the Tables 2-8.

Reviewer 2 Report

Comments and Suggestions for Authors

This review “Therapeutic potential of bovine milk-derived extracellular vesicles”, is interesting, and timely. Here are some highlights and suggestions for the manuscript:

Highlights:

  1. The abstract effectively summarizes the importance of milk extracellular vesicles (MEVs) in human health and their potential therapeutic applications.
  2. The introduction provides a comprehensive background on MEVs, their types, and the relevance of bovine MEVs in particular, setting the stage for the review.
  3. The review covers various aspects of MEV research, including isolation methods, therapeutic potential, and specific applications in gut health, skin and wound healing, antioxidant properties, anti-inflammatory and immuno-modulatory effects, musculoskeletal health, antifibrotic activity, and anticancer properties.

The review discusses the challenges of using MEVs as therapeutic agents, including the lack of a standardized isolation method and the need for more clinical trials.

Suggestions/Drawbacks:

  1. Abstract:
    • Include a statement on the specific aims/objectives of the review.
    • Add a sentence on the implications of the findings for future research or clinical practice.
  2. Introduction:
    • Provide a more structured overview of the sections covered in the review.
    • Clarify the relevance of bovine MEVs in the context of human health and disease.
  3. Isolation and enrichment of bovine MEVs:
    • Discuss the potential impact of different isolation methods on the integrity and function of MEVs.
    • Include a brief comparison of the advantages and disadvantages of each isolation method.
  4. Therapeutic aspects of MEVs:
    • Ensure clarity and organization in discussing the different therapeutic aspects.
    • Provide more context on the specific mechanisms through which MEVs exert their therapeutic effects.
    • Consider discussing the potential challenges or limitations of using MEVs in therapeutic applications.
    • Incorporate recent developments or studies that may have significant implications for MEV-based therapies.
  5. Overall:

o   The review could benefit from more critical analysis and discussion of the limitations of existing studies, such as sample size, study design, and potential biases.

    • Consider discussing the potential future directions and research areas that could advance the field of MEV-based therapeutics.
    • The review could include a section on the potential risks and safety considerations of using MEVs as therapeutic agents, including immunogenicity and off-target effects.
    • The review could discuss the ethical implications of using animal-derived MEVs, particularly in the context of alternative sources or synthetic alternatives.

Overall, the manuscript provides a thorough overview of the therapeutic potential of bovine MEVs, but some areas could be further developed or clarified to enhance its quality and impact.

Author Response

Highlights:

  1. The abstract effectively summarizes the importance of milk extracellular vesicles (MEVs) in human health and their potential therapeutic applications.
  2. The introduction provides a comprehensive background on MEVs, their types, and the relevance of bovine MEVs in particular, setting the stage for the review.
  3. The review covers various aspects of MEV research, including isolation methods, therapeutic potential, and specific applications in gut health, skin and wound healing, antioxidant properties, anti-inflammatory and immuno-modulatory effects, musculoskeletal health, antifibrotic activity, and anticancer properties.

The review discusses the challenges of using MEVs as therapeutic agents, including the lack of a standardized isolation method and the need for more clinical trials.

 Suggestions/Drawbacks:

  1. Abstract:
    • Include a statement on the specific aims/objectives of the review.

As suggested by the reviewer, the abstract was modified stating the specific aims in lines 27-29. 

  • Add a sentence on the implications of the findings for future research or clinical practice.

A sentence was included at the end of the abstract as suggested by the Reviewer. Please see the lines 29-32.

  1. Introduction:
    • Provide a more structured overview of the sections covered in the review.

As suggested by the reviewer, the introduction section was modified with a clear and concise overview of the sub-topics discussed in the latter part of the review. See the lines 70-77.

  • Clarify the relevance of bovine MEVs in the context of human health and disease.

The relevance of bovine MEVs in human health and disease context is included in the introduction. See the lines 54-60.

  1. Isolation and enrichment of bovine MEVs:
    • Discuss the potential impact of different isolation methods on the integrity and function of MEVs.

The effect of EV isolation method on the integrity and function of the MEVs is discussed in lines 136-141.

  • Include a brief comparison of the advantages and disadvantages of each isolation method.

A brief comparison of advantages and disadvantages of each isolation each method is included in the lines 142-153. A summary is also included as the table 1.

  1. Therapeutic aspects of MEVs:
    • Ensure clarity and organization in discussing the different therapeutic aspects.

We would like to acknowledge the comment of the reviewer and a summary of the supportive evidence of each therapeutic effect is included now in a table at the end of each paragraph to improve the clarity. (Table 2-8)

  • Provide more context on the specific mechanisms through which MEVs exert their therapeutic effects.

The Reviewer's concern regarding the in-depth analysis of mechanisms involved in the specific therapeutic potential of MEVs is highly appreciated. However, some of the research articles cited were based on preliminary findings, and no specific studies on the mechanisms and the molecules involved was described as this is an emerging area. Therefore, this review reports only the mechanisms elucidated in the reference articles and we have highlighted the need of such in depth knowledge as future research perspectives in the lines 487-490, 515-518.

  • Consider discussing the potential challenges or limitations of using MEVs in therapeutic applications.

The chapter 4 which is focussed on the challenges of using MEVs as therapeutic agents was improved as suggested and discussed in detail from lines 419-505.

  • Incorporate recent developments or studies that may have significant implications for MEV-based therapies.

Recent publications related to MEVs are included as reference numbers 36,107.

  1. Overall:

The review could benefit from more critical analysis and discussion of the limitations of existing studies, such as sample size, study design, and potential biases.

Thank you very much for highlighting these points. We have improved the content adding further information wherever we could retrieve from the literature (lines 173-175, lines 487-489, line 492-495, lines 515-517).

  • Consider discussing the potential future directions and research areas that could advance the field of MEV-based therapeutics.

A subsection discussing the future directions of the EV research field is included as suggested by the reviewer. (lines 507-518).

  • The review could include a section on the potential risks and safety considerations of using MEVs as therapeutic agents, including immunogenicity and off-target effects.

A subsection 4.4 describing the potential risks and safety considerations is included under section 4 “Challenges of using MEVs as therapeutic agents”. (lines 477-496).

  • The review could discuss the ethical implications of using animal-derived MEVs, particularly in the context of alternative sources or synthetic alternatives.

A sub-section 4.5 describing the ethical implications of using animal-derived MEVs was included under section 4. Challenges of using MEVs as therapeutic agents. (lines 598-505).

Round 2

Reviewer 2 Report

Comments and Suggestions for Authors

I appreciate authors for their efforts in addressing all the comments appropriately. I have no further comments on this manuscript.